# Hypermobile Ehlers–Danlos Syndrome: Diagnostic Challenges and the Role of Genetic Testing

**DOI:** 10.3390/genes16050530

**Published:** 2025-04-29

**Authors:** Irman Forghani, Julia See, William C. McGonigle

**Affiliations:** 1Department of Clinical Genetics, Mount Sinai Medical Center of Florida, Miami Beach, FL 33140, USA; 2Miller School of Medicine, University of Miami, Miami, FL 33146, USAwcm54@med.miami.edu (W.C.M.)

**Keywords:** hypermobile Ehlers–Danlos syndrome, (hEDS), diagnostic criteria, genetic testing, diagnosis of exclusion

## Abstract

**Background/Objectives**: Hypermobile Ehlers–Danlos syndrome (hEDS) is the most common subtype of Ehlers–Danlos syndromes (EDS), a heterogeneous group of hereditary connective tissue disorders. The hallmark features of hEDS include generalized joint hypermobility (GJH), soft or velvety skin, and persistent joint pain. The molecular etiology of hEDS remains unknown, and diagnosis is primarily clinical. The updated diagnostic criteria for hEDS requires the fulfillment of three criteria: (1) GJH, (2) a combination of musculoskeletal and systemic manifestations consistent with a connective tissue disorder, and (3) the exclusion of alternative diagnoses. However, the exclusion process and the role of genetic testing have not yet been fully refined. **Methods**: This retrospective review utilized data from the Hereditary Connective Tissue Disorders (HCTD) patient registry at the University of Miami, which includes individuals evaluated at the HCTD Clinic using a standardized internal clinical and genetic protocol. We analyzed data from 907 patients referred for hEDS evaluation between June 2019 and December 2022. **Results**: Among these patients, 178 met the 2017 diagnostic criteria for hEDS. Genetic testing identified an alternative or additional diagnosis in 47 of these individuals (26.4%), with clinical implications requiring distinct management strategies. **Conclusions**: These findings underscore the importance of criterion three—exclusion of alternative diagnoses—and highlight the critical yet underutilized role of genetic testing in the assessment of joint hypermobility. Furthermore, the results suggest that hypermobility may present a shared phenotype across a spectrum of disorders, including inflammatory diseases, monogenic syndromes, and chromosomal abnormalities.

## 1. Introduction

Hypermobile Ehlers–Danlos syndrome (hEDS) is the most common subtype of EDS, a group of HCTD generally characterized by skin hyperextensibility, joint hypermobility, and tissue fragility. While most EDS subtypes have been linked to specific genetic mutations, the molecular etiology of hEDS remains unknown [1,2].

In 2017, the diagnostic criteria for EDS were reclassified, introducing a long-overdue revision after nearly two decades. The updated criteria aimed to better define the syndromic nature of hEDS, enhance specificity by accounting for the condition’s variable expressivity, and establish a more exclusive, homogeneous cohort to facilitate research. According to the 2017 International Classification for EDS, the diagnosis of hEDS requires meeting three criteria emphasizing clinical presentation, family history, and exclusion of alternative diagnoses [1].

Criterion 1 uses the age and sex-adjusted Beighton scoring system to assess GJH.Criterion 2 consists of three features (A, B, and C), of which two must be present:
⚬Feature A identifies at least five characteristics of connective tissue disorders.⚬Feature B requires a positive family history, with one or more first-degree relatives independently meeting the hEDS diagnostic criteria.⚬Feature C pertains to musculoskeletal complications of joint laxity, such as chronic pain, joint instability, and recurrent dislocations.
Criterion 3 involves the exclusion of alternative diagnoses by ruling out unusual skin fragility, other heritable and acquired connective tissue disorders, and conditions mimicking hEDS [1].

The 2017 diagnostic criteria have been criticized for being overly stringent and failing to adequately capture the multisystemic nature of hEDS. Studies have shown strong associations between hEDS and various extra-articular manifestations, including functional gastrointestinal and genitourinary disorders, postural orthostatic tachycardia syndrome (POTS), and psychological dysfunction. These symptoms often prompt patients to seek evaluation for hEDS [3]. Experts have classified patients who do not completely meet the 2017 classification criteria as having “Hypermobility Spectrum Disorders (HSD)”. However, this classification is not universally recognized in clinical practice [4,5].

Diagnosing hEDS is challenging due to the high clinical variability and the absence of specific biomarkers or a defined molecular cause. Despite efforts to establish a more exclusive diagnostic framework, a structured approach that effectively rules out alternative diagnoses is still lacking. Given this limitation, it is crucial to place greater emphasis on refining and enhancing the role of genetic testing. Currently, genetic testing is underutilized in clinical practice. Many healthcare providers do not routinely incorporate genetic testing into their evaluations, and insurance companies often deny coverage, citing the unknown molecular etiology of hEDS.

Notably, a recent study has introduced 52-kDa fibronectin as a potential biomarker for hEDS [6]. However, the clinical validation and accessibility of this exploratory discovery are still in the early stages. If follow-up studies confirm its applicability, it could provide an objective diagnostic tool that significantly enhances diagnostic precision. Until then, hEDS continues to be a diagnosis of exclusion, requiring a comprehensive evaluation to rule out other heritable and acquired connective tissue disorders as well as other alternative diagnoses.

In this study, we reviewed the HCTD patient registry data for patients referred to the HCTD Clinic at the University of Miami for suspected hEDS between June 2019 and December 2022. All patients underwent evaluation following a standardized internal clinical and genetic protocol. Our findings revealed a considerable number of patients with additional or alternative diagnoses that may have been missed without genetic testing. This study underscores the clinical relevance of criterion three of the 2017 hEDS diagnostic criteria and highlights the role of genetic testing as an essential component of the diagnostic evaluation. Genetic testing can significantly improve diagnostic accuracy, leading to more tailored management strategies.

## 2. Materials and Methods

### Study Design and Population

This retrospective review utilized data extracted from the HCTD patient registry at the University of Miami, covering the period from June 2019 to December 2022. The study included all individuals recorded in the registry who sought evaluation for hEDS and met the 2017 diagnostic criteria for hEDS within this period, as seen in Figure 1.

The HCTD registry includes data from patients evaluated at the HCTD Clinic at the University of Miami, where all individuals were assessed following a standardized internal clinical and genetic evaluation protocol. All patients included in this registry received a diagnosis following a comprehensive clinical evaluation conducted by a clinical geneticist and underwent next-generation sequencing (NGS) using a HCTD gene panel, performed by CLIA-certified clinical genetic laboratories. Patients with low serum alkaline phosphatase levels and a history of osteoporosis, bone fragility, multiple fractures, and dental issues underwent an additional skeletal disorder panel. Those with a history of recurrent infections, urticaria, and multiple allergies and a history of an anaphylactic reaction received panel testing for primary immunodeficiencies, autoimmunity, and autoinflammatory disorders. Patients with a history of episodic abdominal pain and periodic fever were tested for periodic fever syndromes.

Additionally, all patients had their serum tryptase levels checked; those with elevated serum tryptase levels above 11 ng/mL underwent genetic testing for hereditary alpha tryptasemia using the Tryptase Copy Number Variation (CNV) Test from Gene-by-Gene laboratory, which is designed to identify individuals with multiple copies of the α-tryptase gene (*TPSAB1*). Karyotyping, chromosomal microarray, and whole exome or genome sequencing were offered to patients with a history of developmental delay, facial dysmorphism, and recurrent miscarriages and a personal or family history of neurological disorders. This study presents our findings from the registry, including the percentage of patients diagnosed with an additional condition beyond hEDS based on genetic testing results, as well as the specific outcomes of those genetic tests.

## 3. Results

Out of a total of 907 patients referred for suspected hEDS, 178 individuals (19.62%) fulfilled the diagnostic criteria based on clinical evaluation. Notably, among those who met the diagnostic criteria for hEDS, 47 individuals (26.4%) were found to have additional or alternative diagnoses that required distinct management strategies, as indicated by their genetic test results, as listed in Table 1. Our findings revealed that approximately one in four patients who initially met the hEDS diagnostic criteria had an alternative or coexisting condition.

## 4. Discussion

Since the revision of the diagnostic criteria for hEDS in 2017, there has been a notable increase in awareness among healthcare providers and patients. This has led to greater resource allocation and a rise in patients seeking evaluation for joint hypermobility disorders. However, despite the progress made, diagnosing hEDS remains challenging due to its unclear pathogenesis, undefined molecular etiology, variable clinical presentation, and nonspecific multisystem involvement.

The 2017 diagnostic criteria aimed to improve diagnostic specificity by mandating exclusion of differential diagnoses (criterion three). However, it does not provide clear guidance on how to implement this exclusion in a clinical setting. Our analysis of 178 patients who met the hEDS diagnostic criteria revealed that approximately one out of four (26.4%) had additional or alternative diagnoses, which may have been overlooked without genetic testing. These findings suggest a gap in current diagnostic approaches and underscore the critical role of genetic testing in improving diagnostic accuracy.

### 4.1. The Role of Genetic Testing in hEDS Diagnosis

Advances in sequencing technology have enabled the early detection of rare conditions that might otherwise remain undiagnosed. Generalized joint hypermobility is a shared feature in many inherited disorders, and multisystem manifestations of hEDS are usually non-specific and shared across many genetic and multifactorial disorders. The primary use of genetic testing in hEDS evaluation is to exclude alternative diagnoses. This is particularly apparent in distinguishing hEDS from conditions with variable expressivity and age-dependent penetrance. Identifying these conditions based on clinical features alone is difficult, especially in the early stages of the disease when phenotypic manifestations have not fully emerged. However, genetic testing is underused in clinical practice because current clinical guidelines do not endorse routine genetic testing for hEDS, and insurance providers deny coverage, deeming it medically unnecessary. Future revisions of the diagnostic criteria should place greater emphasis on standardizing diagnostic procedures, including the appropriate use of genetic testing.

Our findings suggest that joint hypermobility is a shared phenotype across a spectrum of disorders, including the following:Connective Tissue and Skeletal Disorders—Many of these monogenic disorders present with joint laxity. The presence of a personal or family history suggestive of these conditions, such as aortic disease, sudden cardiac death, retinal detachment, keratoconus, brittle bones, or significant dental anomalies, should prompt evaluation for monogenic skeletal or connective tissue disorders.Neuromuscular Disorders—Hypermobility of the joints and neuromuscular disorders are often interconnected. Joint laxity has been reported as a primary or secondary manifestation in disorders, such as RYR1- and SEPN1-related myopathies, emphasizing the need for genetic testing in addition to neurological assessment in these patients [7].Chromosomal Abnormalities—Joint laxity is a known feature in many chromosomal abnormalities such as triple X syndrome, 1q21.1 deletion syndrome, and 16p13.11 microduplication syndrome [8,9,10,11]. The American College of Medical Genetics and Genomics (ACMG) practice guidelines recommend chromosomal microarray analysis (CMA) as a first-tier test, along with comprehensive genetic testing—such as exome or genome sequencing—when evaluating individuals with developmental delays, intellectual disability, and multiple congenital anomalies [12,13]. Comprehensive genetic testing should also be considered in patients whose clinical course progresses despite standard treatment interventions. This approach may uncover genetic etiologies that could inform prognosis and guide targeted management strategies.Inflammatory Disorders—Studies have shown that pro-inflammatory cytokines (IL-1β, IL-6, IL-8) can compromise ligament integrity in inflammatory conditions [14]. In our cohort, a significant number of patients had underlying inflammatory or immunological disorders such as common variable immunodeficiency type 2 (CVID2) and hereditary α tryptasemia. Hereditary alpha tryptasemia (HαT) is an autosomal dominant condition with an estimated prevalence of 5.5–8% in the general population, caused by increased germline copies of *TPSAB1*. The individuals with extra copies of *TPSAB1* have elevated serum α tryptase and present with multisystem symptoms resembling hEDS, including gastrointestinal dysmotility, autonomic dysfunction, and joint laxity [15]. Given the relatively high prevalence of HAT and significant phenotypic overlap, baseline serum tryptase measurement should be incorporated into the initial diagnostic workup for patients with suspected hEDS. If tryptase levels are elevated, confirmatory genetic testing for *TPSAB1* copy number variation should be considered.

### 4.2. Broadening the Diagnostic Lens: Addressing Multisystemic Features in hEDS

A significant proportion of hEDS patients present with extra-articular manifestations. This multisystem involvement can include functional gastrointestinal disorders, POTS, chronic pelvic pain, and psychological dysfunction. These symptoms are not only frequently observed in hEDS but often constitute the main reason patients pursue an hEDS evaluation. A significant limitation of current diagnostic criteria is their inadequate consideration of these symptoms, which undermine the systemic nature of the disorder. Conditions such as POTS and gastroparesis are excluded because of their nonspecific nature, while other non-specific features such as piezogenic papules and widespread pain are included in the existing criteria. Accordingly, there is a pressing need for more comprehensive diagnostic criteria that encompass the full spectrum of hEDS-related manifestations [1,7,16].

Moreover, the stringent nature of the 2017 criteria inadvertently excludes some patients who do not meet all diagnostic requirements, classifying them under hypermobility spectrum disorders (HSDs)—a designation that lacks universal recognition and may lead to inconsistent management. Additional studies on this subgroup are imperative until emerging diagnostic biomarkers become widely available.

### 4.3. Emerging Biomarkers and the Continued Need for Genetic Testing

Recent research has identified a 52-kDa fibronectin biomarker as a potential diagnostic tool for hEDS [6]. Although this discovery is promising and may represent a step toward the development of an objective diagnostic marker, it is still exploratory at this stage. Clinical validation studies and replication in larger, independent cohorts is necessary to confirm its diagnostic utility and generalizability. Until this biomarker is fully validated and integrated into clinical practice, hEDS will continue to be a diagnosis of exclusion. This necessitates a comprehensive evaluation to rule out other heritable and acquired conditions that mimic hEDS. In the interim, genetic testing will remain an essential tool in refining differential diagnoses, avoiding misclassification, and guiding appropriate management strategies.

### 4.4. A Systematic Diagnostic Approach to hEDS

Given the diagnostic complexity of hEDS, we propose a structured diagnostic approach beginning with a detailed medical history and physical exam, followed by a focused workup guided by a patient’s primary symptoms. For instance, in patients presenting with chronic fatigue—a common but non-specific symptom in hEDS—the diagnostic process should extend beyond the connective tissue disorder itself. The American Academy of Family Physicians (AAFP) Foundation provides invaluable resources for the diagnostic approach and management of fatigue, including a thorough evaluation for metabolic, infectious, or autoimmune etiologies [17,18]. Further diagnostic workup should be tailored to each patient’s clinical presentation, guided by organ-specific findings, and aligned with relevant established clinical guidelines. Specific attention should be directed toward the presence of severe or atypical features, particularly neuromuscular or cardiovascular symptoms, as these findings may necessitate targeted or comprehensive genetic testing. Neuromuscular symptoms should prompt electromyography and nerve conduction studies (EMG/NCV), along with consideration of genetic panels targeting hereditary neuropathies or myopathies [7]. In patients presenting with cardiovascular symptoms and positive family history, genetic screening for arrhythmia and cardiomyopathy-associated genes may be warranted [19,20]. Additionally, in individuals with a history of recurrent pregnancy loss, history of developmental delay, or intellectual disabilities, karyotyping and CMA should be considered to exclude underlying chromosomal abnormalities [12,13,21].

Furthermore, frequent follow-up is indicated in all patients with a diagnosis of hEDS. Symptoms of hEDS are often manageable with appropriate interventions, including injury prevention strategies and targeted treatment of associated comorbidities. However, clinical deterioration despite optimal management should prompt consideration of alternative or coexisting diagnoses. Such scenarios warrant a thorough re-evaluation and may indicate the need for comprehensive genetic testing to identify other underlying or overlapping conditions.

### 4.5. Limitations

This study has several limitations inherent to its retrospective design. First, it was conducted at a single specialized clinic, which may introduce referral and selection bias, as patients seen here are more likely to have complex presentations or previous diagnostic uncertainty, potentially inflating the proportion of individuals with alternative or additional diagnoses. Second, although data were derived from a standardized clinical and genetic evaluation protocol as part of the HCTD registry, genetic testing was not uniformly applied to all participants; rather, it was guided by clinical features and phenotypic presentation. This targeted approach, while practical and cost-effective, may have missed underlying conditions in asymptomatic or atypically presenting individuals. Third, despite the inclusion of multiple gene panels and exome/genome testing in selected cases, not all known or novel pathogenic variants may have been detected, as limitations in variant interpretation and gene coverage still exist even with current next-generation sequencing technologies. Lastly, the absence of longitudinal follow-up data limits our ability to assess the clinical impact of these additional diagnoses on long-term patient outcomes.

Future prospective studies incorporating universal testing protocols and broader multi-institutional cohorts are warranted to validate these findings and further delineate the diagnostic overlap between hEDS and other genetic or immunologic disorders.

## 5. Conclusions

The 2017 revised criteria for hEDS and emerging potential diagnostic biomarkers are significant steps toward standardizing the diagnosis of hEDS. However, there is still a need for ongoing refinement and validation of these new developments. Diagnosing complex medical conditions, such as hEDS, still heavily relies on clinical assessment and careful elimination of other potential cause(s). Criterion three, which focuses on excluding alternative diagnoses, is a crucial component of the diagnostic criteria and should be emphasized in clinical practice. Our findings particularly underscore the need for incorporating genetic testing in criterion three, as it plays a pivotal role in improving diagnostic accuracy and identifying coexisting or alternative heritable disorders.

To enhance diagnostic rigor and prevent premature closure, a systematic, step-by-step diagnostic approach supported by evidence-based medicine is essential. Clinicians should be especially vigilant when assessing patients with significant neurologic, musculoskeletal, or cardiovascular symptoms, as these may indicate conditions beyond hEDS. Long-term follow-up and comprehensive genetic testing should be pursued when indicated.

Beyond its diagnostic applications, genetic testing has the potential to inform and reshape future revisions of the hEDS diagnostic criteria, particularly as insights into the condition’s molecular basis continue to emerge. Additionally, a multidisciplinary approach—bringing together expertise in genetics, cardiology, neurology, rheumatology, and allied health disciplines such as physiotherapy, occupational therapy, and psychological support—is essential for delivering comprehensive, coordinated care. This collaborative model ensures that the diverse and multisystemic manifestations of hEDS are thoroughly evaluated and managed, leading to more personalized treatment plans and improved patient outcomes.

Finally, continued research into both the genetic architecture and clinical progression of hEDS is critical for improving diagnostic precision, guiding management strategies, and optimizing long-term outcomes for affected individuals.

## Figures and Tables

**Figure 1 genes-16-00530-f001:**
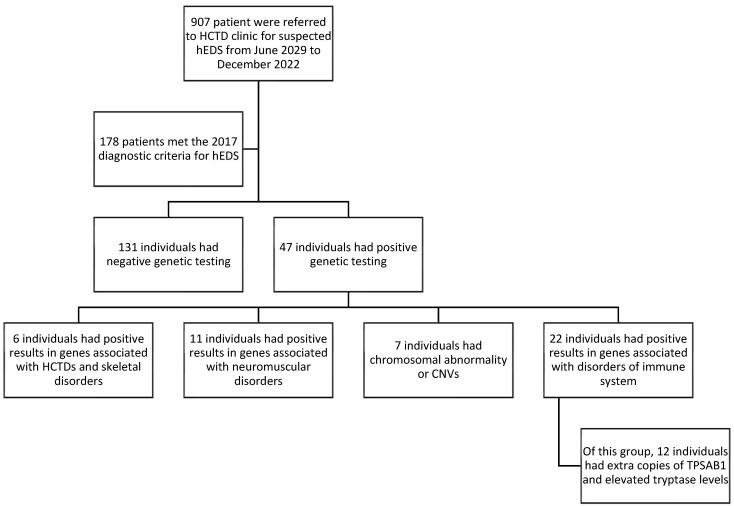
Patient inclusion flowchart for hEDS evaluations in the HCTD registry at the University of Miami (June 2019–December 2022).

**Table 1 genes-16-00530-t001:** Patients meeting the 2017 hEDS criteria with alternative or coexisting diagnoses identified by genetic testing.

Age at Diagnosis (Year)/Sex	Genetic Test Results; Classification; Zygosity	Associated Diagnosis
Inherited connective tissue and skeletal disorders		
49/F	NM_005902.4(SMAD3):c.770_771del (p.Val257fs); PV; Het	LDS, type 3 (OMIM #613795)
35/F	NM_001457.4(FLNB):c.4514+1G>A; LPV; Het	Larsen syndrome (OMIM #150250)
31/M	NM_000501.4(ELN):c.542-2A>T; LPV; Het	Cutis Laxa (OMIM #123700)
56/F	NM_000478.6(ALPL):c.1417G>A (p.Gly473Ser); PV; Het	HPP, adult-onset (OMIM #146300)
19/F	NM_000478.6(ALPL):c.407G>A (p.Arg136His); PV; Het	HPP, adult-onset (OMIM #146300)
46/F	NM_001853.4(COL9A3):c.1739dup (p.Gly581fs); PV/LPV; Het	Epiphyseal dysplasia, multiple, 3 (OMIM #600969)
Neuromuscular disorders		
25/F	NM_000083.3(CLCN1):c.1238T>G (p.Phe413Cys); PV/LPV; Het	Myotonia congenital (OMIM #160800)
49/F	NM_001849.4(COL6A2):c.2611G>A (p.Asp871Asn); PV/LPV; Het	Collagen VI-Related Myopathy (OMIM #620725)
44/M	NM_000080.4(CHRNE):c.764C>T (p.Ser255Leu); PV/LPV; Het	SCCMS (OMIM #605809)
53/F	NM_000070.3(CAPN3):c.1303G>A (p.Glu435Lys); PV/LPV; Het	LGMDD4 (OMIM #618129)
69/F	NM_002693.3(POLG):c.1760C>T (p.Pro587Leu); PV/LPV; Het	POLG-related spectrum disorder (OMIM #157640)
61/F	NM_002693.3(POLG):c.2243G>C (p.Trp748Ser); PV/LPV; Het	POLG-related spectrum disorder (OMIM #157640)
32/F	NM_002693.3(POLG):c.1399G>A (p.Ala467Thr); PV; Heterozygous ATXN8OS:CTG[156] Repeat expansion; PV; Het NM_053274.3(GLMN):c.554_558delinsG (p.Lys185fs); PV; Het	POLG-related spectrum disorder (OMIM #157640) SCA type 8 (OMIM #608768) Glomuvenous malformation (OMIM #138000)
43/F	PABPN1: GCG[11] Alanine repeat expansion; LPV; Het	Oropharyngeal muscular dystrophy (OMIM #164300)
26/F	ATXN8OS: CTG[101] Repeat Expansion; PV; Het	SCA type 8 (OMIM #608768)
37/F	ATXN10: ATTCT[2011] Repeat Expansion; PV; Het	SCA type 10 (OMIM #603516)
51/F	ATXN10: ATTCT[793] Repeat Expansion; PV; Het	SCA type 10 (OMIM #603516)
Chromosomal abnormalities and copy number variations		
16/F	NC_000016.10:g.(28796097_28796872)_(29032542_29032651)dup	16p11.2 duplication syndrome (OMIM #614671)
28/F	arr[GRCh37] 16p13.11(15493046_16362651)x3 NC_000016.10:g.15493046_16362651dup	16p13.11 microduplication syndrome (ORPHA:261243)
26/F	seq[GRCh38] del(15)(q13.2q13.3)pat NC_000015.10:g.(30,066,060_30,622,982)_(32,149,374_32,632,609)del	15q13.3 deletion syndrome (OMIM #612001)
14/F	arr[GRCh37] 1q21.1(144893419_145888926)x1 NC_000001.10:g.(144893419_145888926)del	1q21.1 deletion syndrome (IMIM #612474)
49/F	Karyotype: monosomy X in 24% of the reviewed metaphase cells (12/50)	Mosaic Turner syndrome (ORPHA:881)
32/F	Karyotype: 47, XXX NM_004415.4(DSP):c.4531C>T (p.Gln1511Ter); PV; Het	Triple X syndrome (ORPHA:3375) ARVD8 (OMIM #607450)
31/F	Karyotype: 47, XXX	Triple X syndrome (ORPHA:3375)
Disorders of the Immune System		
36/F	NM_012452.3(TNFRSF13B):c.542C>A (p.Ala181Glu); PV/LPV; Het	CVID2 (OMIM #240500)
49/F	NM_012452.3(TNFRSF13B):c.542C>A (p.Ala181Glu); PV/LPV; Het	CVID2 (OMIM #240500)
18/F	NM_012452.3(TNFRSF13B):c.204dup (p.Leu69fs); PV/LPV; Het	CVID2 (OMIM #240500)
58/F	NM_012452.3(TNFRSF13B):c.204dup (p.Leu69fs); PV/LPV; Het	CVID2 (OMIM #240500)
42/F	NM_012452.3(TNFRSF13B):c.542C>A (p.Ala181Glu); PV/LPV; Het	CVID2 (OMIM #240500)
40/F	NM_012452.3(TNFRSF13B):c.310T>C (p.Cys104Arg); PV/LPV; Het	CVID2 (OMIM #240500)
51/F	NM_012452.3(TNFRSF13B):c.310T>C (p.Cys104Arg); PV/LPV; Het NM_003900.5(SQSTM1):c.1175C>T (p.Pro392Leu); PV/LPV; Het	CVID2 (OMIM #240500) SQSTM1-related condition (OMIM #617158)
21/F	NM_000243.3(MEFV):c.2177T>C (p.Val726Ala); PV/LPV; Het	FMF, AD (OMIM #134610)
20/F	NM_000383.4(AIRE):c.769C>T (p.Arg257Ter); PV; Het	APS1 (OMIM #240300)
48/F	NM_001122764.3(PPOX):c.1092_1093del (p.Arg364fs); PV; Het	Porphyria variegata (OMIM #176200)
59/F 54/F 54/F 49/M 46/F 21/F 51/F 53/F 11/M 74/F 46/M 28/F	Increased copies of TPSAB1	HαT
Metabolic disorders		
42/F	NM_000410.4(HFE):c.187C>G (p.His63Asp); PV/LPV; Het NM_000410.4(HFE):c.845G>A (p.Cys282Tyr); PV/LPV; Het	Hemochromatosis type 1 (OMIM #235200)

APS1: autoimmune polyendocrinopathy syndrome type I; ARVD8: arrhythmogenic right ventricular dysplasia-8; CVID2: common variable immunodeficiency, type 2; FMF, AD: familial Mediterranean fever, autosomal dominant; HαT: hereditary alpha tryptasemia; Het: heterozygous; HPP: hypophosphatasia; LDS: Loeys–Dietz syndrome; LGMDD4: autosomal dominant limb-girdle muscular dystrophy-4; LPV: likely pathogenic variant; PV: pathogenic variant; SCA: spinocerebellar ataxia; SCCMS: slow-channel congenital myasthenic syndrome.

## Data Availability

The raw data supporting the conclusions of this article will be made available by the authors upon request.

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
