# Peer review of "Hypermobile Ehlers–Danlos Syndrome: Diagnostic Challenges and the Role of Genetic Testing"

_genes, 2025, doi:10.3390/genes16050530_

Round 1
Reviewer 1 Report
Comments and Suggestions for Authors
Accurately diagnosing Hypermobile Ehlers-Danlos Syndrome (hEDS) is challenging due to the lack of known genetic markers, some symptoms overlap with other conditions, therefore, it is of great importance to refine the testing process. In this paper, the authors described their findings on the role of genetic testing in criterion three for diagnosing hEDS. Among 907 patients, ~20% fulfilled the diagnostic criteria based on the clinical evaluation, and among them, almost one out of four had additional/alternative diagnoses. This study highlighted the importance of genetic testing process in the diagnosis of hEDS. However, there are some limitations with the current study which are also discussed in the paper, such as the limited scope of patients may introduce selection bias, and the lack of long-term follow-up data limits the assessment of clinical impact. Despite these limitations, this paper further emphasized the crucial role of incorporating other testing techniques such as genetic testing in criterion three for diagnosing hEDS.
Author Response
Comment 1: Accurately diagnosing Hypermobile Ehlers-Danlos Syndrome (hEDS) is challenging due to the lack of known genetic markers, some symptoms overlap with other conditions, therefore, it is of great importance to refine the testing process. In this paper, the authors described their findings on the role of genetic testing in criterion three for diagnosing hEDS. Among 907 patients, ~20% fulfilled the diagnostic criteria based on the clinical evaluation, and among them, almost one out of four had additional/alternative diagnoses. This study highlighted the importance of genetic testing process in the diagnosis of hEDS. However, there are some limitations with the current study which are also discussed in the paper, such as the limited scope of patients may introduce selection bias, and the lack of long-term follow-up data limits the assessment of clinical impact. Despite these limitations, this paper further emphasized the crucial role of incorporating other testing techniques such as genetic testing in criterion three for diagnosing hEDS.
Response 1: We thank the reviewer for their thoughtful and constructive comments. We appreciate the recognition of the importance of refining the diagnostic process for hEDS and the potential role of genetic testing in criterion three. As noted, the limitations regarding the potential for selection bias due to the patient population and the lack of long-term follow-up data have been explicitly discussed in the manuscript. These were important considerations in our interpretation of the findings, and we agree that they warrant continued investigation in future studies with broader and longitudinal cohorts. We have reviewed the relevant sections to ensure that these limitations are clearly articulated and appropriately contextualized.
Reviewer 2 Report
Comments and Suggestions for Authors
Thank you for the opportunity to review your manuscript, "Hypermobile Ehlers-Danlos Syndrome: Diagnostic Challenges and The Role of Genetic Testing." This is a valuable and timely study that addresses a critical gap in the diagnostic process for hEDS. By focusing on the underutilised Criterion 3 of the 2017 diagnostic framework, your work makes a strong case for integrating genetic testing to exclude other heritable and clinically overlapping disorders.
Some suggestions:
1. While the manuscript is well-written, some long and complex sentences could be simplified to improve readability. A light round of language editing is recommended.
- Table 1 is informative but dense. Consider breaking it down into thematic categories (e.g., neuromuscular, chromosomal, connective tissue disorders) or presenting the most frequent diagnoses separately for clarity.
- A visual summary—such as a diagnostic algorithm or decision tree—would enhance the accessibility and clinical utility of your proposed diagnostic approach.
- In the discussion of treatment implications or emerging biomarkers (e.g., fibronectin), clearly distinguish between exploratory findings and those supported by larger or replicated studies.
- Consider expanding the conclusion slightly to reflect broader implications, such as the potential role of genetic testing in reshaping future hEDS diagnostic criteria or the importance of multidisciplinary evaluation.
The manuscript is generally well-written, with a clear and professional tone. The scientific content is conveyed effectively, and the overall structure supports comprehension. However, there are a few areas where minor language improvements would enhance readability and flow.
Some sentences are lengthy or complex and could benefit from simplification or restructuring. Additionally, transitions between paragraphs or sections (particularly in the discussion) could be made smoother to improve cohesion.
A light round of language editing is recommended to refine grammar, punctuation, and sentence clarity. These changes are not essential for understanding but would polish the manuscript for publication and improve accessibility to a broader readership.
Author Response
Response to Reviewer 2:
We sincerely thank the reviewer for their thoughtful and constructive feedback. We have carefully considered each of the suggestions and made the following revisions to improve the clarity, structure, and impact of our manuscript:
Comment 1: While the manuscript is well-written, some long and complex sentences could be simplified to improve readability. A light round of language editing is recommended.
Response 1: We appreciate the comment regarding sentence complexity. The manuscript has been reviewed and revised for clarity and readability, with attention to simplifying overly complex sentences while preserving scientific accuracy. We have uploaded the revised version and a clear version for your review.
Comment 2: Table 1 is informative but dense. Consider breaking it down into thematic categories (e.g., neuromuscular, chromosomal, connective tissue disorders) or presenting the most frequent diagnoses separately for clarity.
Response 2: Table 1 has been reorganized into thematic categories, including neuromuscular, chromosomal, and connective tissue disorders, to enhance interpretability. We highlighted each category to increase the contrast and used abbreviations in the table to help with density.
Comment 3: A visual summary—such as a diagnostic algorithm or decision tree—would enhance the accessibility and clinical utility of your proposed diagnostic approach.
Response 3: We appreciate the suggestion to include a visual summary to enhance the clinical utility of our diagnostic approach. In response, we have added an Inclusion Flowchart (Figure 1) to clarify participant selection. However, as the internal protocol utilized in this study has not been externally validated and given that the primary focus of this manuscript is not the diagnostic process itself, we believe it is premature to include a diagnostic algorithm. Introducing such an element at this stage may also detract from the central objective of the paper. A robust, evidence-informed diagnostic framework for suspected hEDS remains a critical area for future investigation, and further comprehensive work is necessary before such an algorithm can be reliably proposed.
Comment 4: In the discussion of treatment implications or emerging biomarkers (e.g., fibronectin), clearly distinguish between exploratory findings and those supported by larger or replicated studies.
Response 4: The discussion on the 52-kDa fibronectin biomarker has been revised to clearly distinguish it as an exploratory finding. We emphasize that although the biomarker shows promise, its clinical utility remains unproven pending validation through larger, independent studies.
Comment 5: Consider expanding the conclusion slightly to reflect broader implications, such as the potential role of genetic testing in reshaping future hEDS diagnostic criteria or the importance of multidisciplinary evaluation.
Response 5: The conclusion has been expanded to reflect the broader implications of our findings. Specifically, we discuss how genetic testing may inform future revisions of the hEDS diagnostic criteria and underscore the importance of a multidisciplinary approach—including genetics, cardiology, neurology, rheumatology, and allied health—for comprehensive patient care.
We believe these revisions have substantially improved the manuscript and aligned it more closely with the reviewer’s insightful recommendations.